# What Shapes a Creative Machine Mind? Comprehensively Benchmarking Creativity in Foundation Models

## Abstract

The meteoric rise of foundation models (FMs) has expanded their capabilities far beyond conventional tasks. Creativity, long regarded as a hallmark of human intelligence and a driver of innovation, is now increasingly recognized as a critical dimension of machine intelligence in the era of generative FMs, complementing traditional measures of accuracy. However, existing evaluation frameworks for creativity remain fragmented, relying on ad hoc metrics that are not firmly grounded in established theories of creativity. To address this gap, we introduce $C^2$-Eval, a holistic benchmark for the unified assessment of creativity in FMs. Specifically, $C^2$-Eval distinguishes between two complementary forms of Creativity ($C^2$): **convergent** creativity, where tasks admit constrained solutions (e.g., code generation), and **divergent** creativity, where tasks are open-ended (e.g., story telling). It evaluates both dimensions using fine-grained assessments derived from social science theories, focusing on Usefulness, Originality, and Surprise (U-O-S). Through extensive experiments on leading proprietary and open-source models, we provide a comprehensive analysis of trade-offs in their creative capabilities. Our results highlight the capabilities and challenges of current FMs in pursuing a creative machine mind, showing that $C^2$-Eval provides an effective lens for examining the evolving landscape of creative AI.

## 1 Introduction

Creativity is a fundamental aspect of human intelligence, widely characterized in the social sciences as the ability to generate content that is useful, original, and surprising (**U-O-S**) (Simonton, 2012a; Amabile, 2018). The rapid development of Foundation Models (FMs), driven by major advances in their reasoning abilities (Guo et al., 2025; Yang et al., 2025; OpenAI, 2025), has led to remarkable performance on a wide range of conventional tasks (Joshi et al., 2017; Chen et al., 2021). As FMs are increasingly positioned as collaborators in science, art, and open-ended problem-solving, only measuring their accuracy becomes fundamentally insufficient (Zhang et al., 2025; Fang et al., 2025), but helping humans find diverse and high-quality solutions to complex problems (Zhao et al., 2025). Developing a rigorous framework to quantify and understand the creative potential of FMs is an essential step for responsibly guiding their development and unlocking their complete power.

Previous studies have investigated the creative potential of FMs in specific domains, such as question answering and coding (He et al., 2025; Lu et al., 2024b; DeLorenzo et al., 2024b), as well as novel story and idea generation (Fang et al., 2025; Stevenson et al., 2022). While valuable, the primary limitation of these efforts is that they provide fragmented evaluations of FMs' creativity w.r.t. both the task and metric. Existing frameworks typically focus on either constrained tasks with objective answers (like code generation) or fully open-ended tasks (like story writing), overlooking the important interplay between different facets of machine creativity. Second, this fragmentation is also mirrored in how creativity itself is measured. Most evaluations assess the **U-O-S** triplet of creativity in isolation, rather than integrating them into a unified framework, failing to provide a reliable and complete understanding of the creative capabilities of modern FMs.

To address these gaps, we build on the theory of creativity in social science (**U-O-S**) (Simonton, 2012b) and introduce $C^2$-Eval, a comprehensive framework to evaluate FM creativity. We first pro-

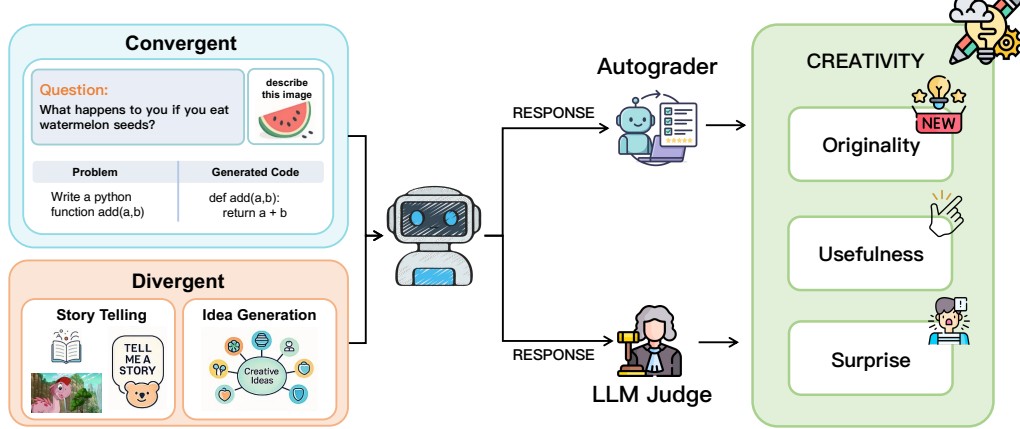

Figure 1: Overview of our C$^2$-Eval framework. **Left panel**: the two distinct creativity regimes: convergent, which focuses on structured tasks like question answering and code generation, and divergent, which evaluates open-ended tasks such as story telling and idea generation. **Right panel**: the **U-O-S** components of creativity used to evaluate responses across both regimes.

pose a novel task-level classification of creativity: **convergent Creativity and divergent Creativity** (**C$^2$**), as illustrated in the left panel of Figure 1. Convergent creativity aims to assess an FM's ability to generate novel and varied solutions within the structured constraints of traditional tasks, such as question answering and code generation. In parallel, we define divergent creativity as the measure of the model's capacity to explore a wide range of possibilities in open-ended scenarios like story telling. Furthermore, in both regimes, every generated response is evaluated through a unified lens composed of the three core **U-O-S** components of creativity, as depicted in the right panel of Figure 1. Compared to previous studies, our C$^2$-Eval framework enables a holistic and rigorous measurement of FMs creativity, capturing its innovative abilities in both structured problem solving and unconstrained generation from diverse perspectives.

Based on our C$^2$-Eval framework, we conduct an extensive evaluation of over 20 leading open-source and proprietary FMs, where we reveal several key insights. Our analysis first maps the current performance landscape, identifying the most creative proprietary and open-source models and highlighting their distinct strengths. Furthermore, our results reveal complex dynamics in model scaling; we find that larger FMs do not always have better creative performance, but that advanced reasoning capabilities appear to provide increasingly creative returns at scale. Our framework also introduces new insights about the nature of machine creativity itself by identifying a nuanced correlation between a model's convergent and divergent creativity abilities. Finally, through ablation studies, we demonstrate the significant positive influence of carefully designed creative instructions on models' output. We summarize our key contributions as follows:

- We introduce C$^2$-Eval, the first comprehensive benchmark designed to systematically evaluate FMs' creativity across both *convergent* and *divergent* task regimes, unifying the assessment of structured tasks with open-ended generations (see Section 3.1).

- We propose a principled evaluation methodology for FMs' creativity grounded in social science. It measures creativity with three classic dimensions: **usefulness, originality, and surprise** (**U-O-S**), providing a consistent evaluation strategy (see Section 3.2).

- We conduct an extensive empirical study of over 20 common closed- and open-source models, establishing a comprehensive landscape for creative AI, revealing nuanced performance trade-offs and providing a robust toolkit for guiding future research (see Section 4.2).

## 2 RELATED WORK

**Creativity in Social Science.** Following classic social–psychological works, creativity is typically defined at the product level as outcomes that are both original and useful/appropriate (Stein, 1953; Runco & Jaeger, 2012). Major reviews further emphasize that judgments of creativity are strongly

contextual and socially mediated (Hennessey & Amabile, 2010). Amabile (1983; 2018), in the componential theory, argue that domain-relevant skills, creativity-relevant processes/strategies, and intrinsic motivation (as shaped by the social environment) jointly determine creative performance; Csikszentmihalyi (1999); Csikszentmihalyi et al. (1997) conceive creativity as the product of person–domain–field interactions, with the field serving as the gatekeeper of what counts as creative. We also draw on the TTCT tradition (e.g., fluency, flexibility, originality, elaboration) for back-grounded characterization (Torrance, 1966; Alabbasi et al., 2022). Building on this consensus, we adopt **U-O-S** as our core foundational evaluation framework which remains compatible with complementary perspectives such as the Four-C model (ranging from everyday to eminent creativity) and the 4P/5A approaches (from person and process to sociocultural ecology) (Kaufman & Beghetto, 2009; Rhodes, 1961; Glăveanu, 2013; Boden, 2004; Simonton, 2012b).

**Evaluating LLM Creativity.** Existing research on LLM creativity largely follows two trajectories: structured tasks and open-ended tasks. In structured tasks, the main practice treats functional correctness as the primary gate and adopts automated, reproducible criteria to implement two-stage protocol 'first correct, then evaluation'. In code generation, multiple samples are drawn to secure a correct answer; creativity is then assessed on the set of qualified outputs along dimensions such as fluency, flexibility, originality, and elaboration (DeLorenzo et al., 2024a). A complementary line of work imposes incremental constraints during generation to elicit distinct solutions (Lu et al., 2024b). In QA settings, once correctness is achieved, similarity metrics are used to quantify the diversity of correct solution types (He et al., 2025).

In contrast, open-ended tasks lack a single ground truth and therefore admit more varied evaluation protocols. In the text-to-text writing-prompts setting, a prompt specifies a scenario or stylistic constraints and models generate short stories, continuations, or copy; evaluation may be performed by human raters or, increasingly, by LLM-as-Judge schemes that score coherence, style, originality, and related dimensions to improve scalability and consistency (Fan et al., 2018; Zheng et al., 2023; Liu et al., 2023; Zhao et al., 2025). In multimodal open-ended evaluation, one approach maps free-form generations to predefined options at test time (Liu et al., 2024); another expands benchmarks into multi-domain, cross-disciplinary challenges to enable more robust and comprehensive assessment (Yue et al., 2024). Classical psychological instruments,such as AUT and TTCT, are also employed to probe divergent creative capacity (Torrance et al., 2008; Stevenson et al., 2022).

In summary, these practices have been instrumental for probing FMs' "creativity," yet most studies remain tailored to a single task or modality and lack an integrated, cross-task synthesis. To address this gap, we build on the **U-O-S** paradigm and propose a unified, systematized evaluation framework that standardizes judging protocols, dataset splits, and metrics across open-ended and verifiable settings, enabling comparable creativity assessments at scale.

## 3 METHODOLOGY

### 3.1 DEFINITION

#### 3.1.1 THE STANDARD DEFINITION OF CREATIVITY

Our framework is grounded in the standard, multi-component view of creativity established in psychological and philosophical research. The cornerstone of this view is the bipartite definition, which posits that a creative product must be both original (novel or unique) and useful (appropriate or effective for its context) (Runco & Jaeger, 2012). Building upon this, subsequent work has incorporated a third, crucial dimension: surprise, which captures the non-obviousness or unexpectedness of an idea, distinguishing true ingenuity from straightforward problem-solving (Simonton, 2012b).

Aligning with these foundational treatments (Paul & Kaufman, 2014), we adopt this three-component framework to define creativity. Concretely, we operationalize creativity via:

- **Usefulness** requires an idea to be effective and practically applicable to the task at hand;
- **Originality** requires an idea to be genuinely new rather than a mere rephrasing of existing solutions;
- **Surprise** captures a solution's non-obviousness beyond what could be achieved through conventional, algorithmic derivations.

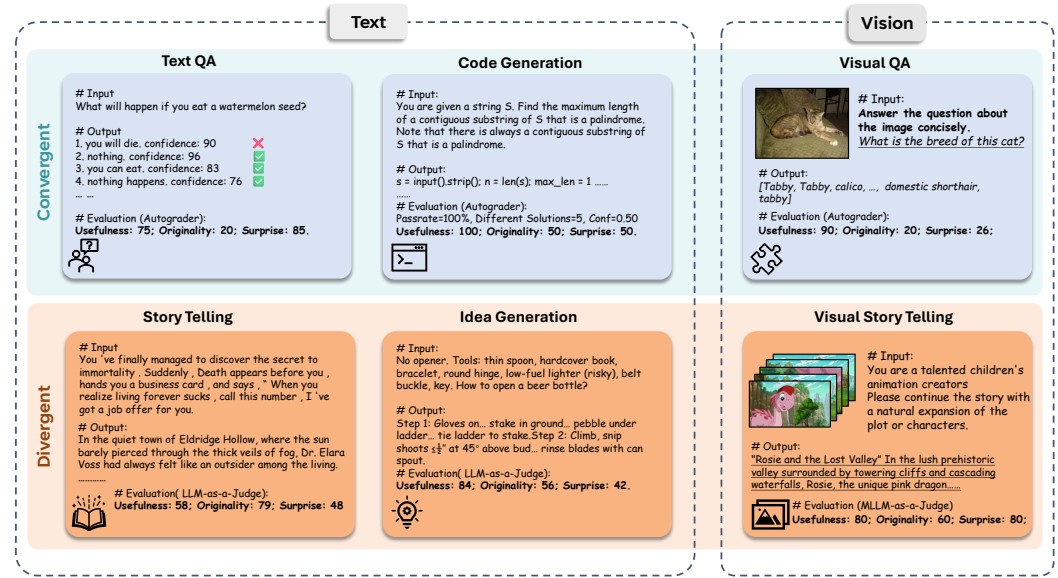

Figure 2: Illustration of Sample evaluation cases within our C$^2$-Eval framework. The *top panel* presents examples of convergent creativity tasks, including Textual QA, Code Generation, and Visual QA, where an Autograder assesses the U-O-S triplet. The *bottom panel* displays divergent creativity tasks, such as Story Telling, Idea Generation, and Visual Story Telling, where an advanced LLM judges responses based on the same U-O-S components. Details of Autograder and LLM-as-a-Judge we used here can be found in Section 3.2.

### 3.1.2 TASK-DEPENDENT OPERATIONALIZATION: CONVERGENT VS. DIVERGENT CREATIVITY

While the U-O-S triplet provides a robust conceptual foundation, its direct application across a diverse benchmark is non-trivial. The core challenge is that the method of measuring each component is fundamentally contingent on the task's objective constraints. A single, unified measurement protocol is insufficient because the very nature of creative expression differs between tasks with objective solutions versus those that are open-ended. To address this, our methodology extends beyond the standard definition by classifying tasks into two regimes, Convergent Creativity and Divergent Creativity, as shown in Figure 2. This classification enabling a more nuanced and context-aware evaluation and dictating how we measure the three core components:

**Convergent Creativity** pertains to tasks characterized by well-defined constraints and objective correctness criteria, where the goal is to converge upon a correct solution set (e.g., factoid QA, code generation, mathematical reasoning). Within this regime, Usefulness is mapped to correctness, Originality to the diversity among correct solutions, and Surprise to the non-obviousness of a given correct answer. Here, creativity often manifests in the problem-solving pathway.

**Divergent Creativity** addresses open-ended tasks lacking a single ground-truth answer, encouraging the exploration of a vast solution space (e.g., unimodal and multimodal story continuation). In this context, where objective correctness is absent, Usefulness is interpreted as appropriateness and coherence, Originality as perceived novelty, and Surprise as unexpectedness. Here, creativity is embodied directly within the generated content itself.

### 3.2 EVALUATION METRICS

#### 3.2.1 CONVERGENT CREATIVITY

The following pipeline illustrates our autograder. For each prompt in the convergent creativity regime, we generate a set of $n$ candidate outputs, denoted by $\{y_i\}_{i=1}^{n}$. To evaluate them, we first introduce an indicator function, $\mathbb{I}(y_i) \in \{0, 1\}$, which returns 1 if the output $y_i$ is correct according to the task's specific constraints, and 0 otherwise.

**Usefulness.** In the convergent regime, usefulness is synonymous with correctness. We therefore define the *usefulness* score, $U$, as the accuracy over the $n$ generated samples: $U = \frac{1}{n} \sum_{i=1}^{n} \mathbb{I}(y_i)$.

**Originality.** We measure the originality of the generated response as the semantic diversity among the set of correct solutions. To this end, we partition the correct outputs into disjoint clusters using a threshold-based rule: any two correct outputs $y_i$ and $y_j$ are assigned to the same cluster if their cosine similarity meets or exceeds a pre-defined threshold $\tau$ (e.g., 0.9). Each problem $p$ yields a count $K_p$ of distinct clusters, and the prompt-level *originality* score is defined as $O = K_p$.

**Surprise.** We define Surprise as the complement of the model's own confidence-of-correctness estimate: lower self-reported confidence implies higher Surprise. Given a prompt $x$, we generate $n$ candidate answers $\{y_i\}_{i=1}^{n}$ via multi-sample sampling. After producing each $y_i$, we then instruct the model to generate a confidence score $c_i \in [0, 1]$ that indicates the probability that $y_i$ is correct. (Xiong et al., 2023) Let $z_i \in \{0, 1\}$ denote the ground-truth correctness of $y_i$. Following Kadavath et al. (2022), the prompt-level confidence averages confidence over correct samples only by

$$C = \frac{\sum_{i=1}^{n} z_i \, c_i}{\max\left(1, \ \sum_{i=1}^{n} z_i\right)}.$$

We then define the prompt-level *surprise* metric as the complement: $S = 1 - C$.

### 3.2.2 DIVERGENT CREATIVITY

For open-ended generation tasks that do not have a single ground-truth answer, we employ an automatic rater, generally the most advanced judge model (Tan et al., 2024), to score each generated sample (Fan et al., 2018; Zheng et al., 2023; Liu et al., 2023; Zhao et al., 2025). To ensure a standardized and rigorous evaluation, the rater is provided with a detailed prompt containing explicit definitions and multi-point scoring rubrics for each of our three core creativity components.

The evaluation proceeds in two stages. First, for a given input prompt, we generate $K_{\text{total}}$ distinct output samples. Each of these samples is then individually evaluated by the automatic rater, yielding a sample-level score triplet $(U_i, O_i, S_i) \in [0, 5]^3$ for each of the $i = 1, \ldots, n$ samples. These scores (the larger the better) reflect the following conceptual underpinnings:

- **Usefulness** ($U$) measures the coherence, logical consistency, and contextual appropriateness of the generated output. It assesses whether the response is a well-structured and meaningful continuation of the prompt, free of contradictions or nonsensical elements.
- **Originality** ($O$) assesses the novelty of the ideas presented. It quantifies the degree to which the output moves beyond clichés, predictable narrative paths, and conventional tropes to introduce fresh, imaginative concepts.
- **Surprise** ($S$) captures the non-obviousness and emotional engagement of the output. It measures the presence of unexpected plot twists, striking imagery, or insightful developments that make the generation compelling and memorable.

Finally, we aggregate these sample-level scores to produce a stable, **prompt-level** score for each metric by taking the average across all samples, as described by the unified aggregation formula: $M_{\text{prompt}} = \frac{1}{n} \sum_{i=1}^{n} M_i, \quad \text{where } M \in \{\mathrm{U}, \mathrm{O}, \mathrm{S}\}$.

**Overall Creativity Score.** For convergent and divergent creativity, we first scale the range of evaluated values into [0, 100], obtaining a triplet $(\mathrm{U}, \mathrm{O}, \mathrm{S}) \in [0, 100]^3$. The score range is selected for convenient comparison and interpretation. We then compute the final composite creativity score, $C$, by taking the average of these three components.

## 4 BENCHMARKING CREATIVITY IN FMS

### 4.1 SETUP

**Models.** To provide a comprehensive assessment of creative capabilities, we evaluate a diverse cohort of both proprietary and open-source models across text-only and vision-language paradigms.

Our evaluation of proprietary models encompasses recent releases from OpenAI and Anthropic. From OpenAI, we include GPT-4o, GPT-4o-mini, and o4-mini model and both of GPT-4o-mini and

| Model Name | Combined Creativity | Convergent Creativity | | | | | | | | Divergent Creativity | | | | | | | |
| --- | --- | --- | --- | --- | --- | --- | --- | --- | --- | --- | --- | --- | --- | --- | --- | --- | --- |
| | | Question Answering | | | | Code Generation | | | | Story Telling | | | | Idea Generation | | | |
| | | Use. | Orig. | Surp. | Crea. | Use. | Orig. | Surp. | Crea. | Use. | Orig. | Surp. | Crea. | Use. | Orig. | Surp. | Crea. |
| Qwen2.5-7B | 31.9 | 52.8 | 13.9 | 19.5 | 28.7 | 22.7 | 30.0 | 25.5 | 26.1 | 52.7 | 53.4 | 39.9 | 48.7 | 25.3 | 22.3 | 15.0 | 23.9 |
| Qwen2.5-14B | 35.0 | 67.6 | 13.1 | 11.8 | 30.8 | 43.9 | 36.8 | 31.0 | 37.2 | 55.1 | 61.9 | 40.5 | 52.5 | 21.1 | 17.1 | 11.8 | 19.5 |
| Qwen2.5-32B | 35.4 | 66.7 | 6.2 | 16.2 | 29.7 | 48.5 | 49.4 | 14.7 | 37.5 | 48.6 | 63.4 | 34.3 | 48.8 | 26.4 | 24.5 | 17.5 | 25.7 |
| Qwen2.5-72B | 35.3 | 75.5 | 7.9 | 6.7 | 30.0 | 56.6 | 43.4 | 15.2 | 38.4 | 44.7 | 66.2 | 31.6 | 47.5 | 27.2 | 21.4 | 15.3 | 25.1 |
| *Qwen3-8B | 40.2 | 59.7 | 8.4 | 17.5 | 28.5 | 84.2 | 70.2 | 20.8 | 58.4 | 66.8 | 71.0 | 57.4 | 65.1 | 29.4 | 31.1 | 21.2 | 29.4 |
| *Qwen3-235B | 58.3 | 86.6 | 12.2 | 9.6 | 36.1 | 91.2 | 78.4 | 22.8 | 64.1 | 85.5 | 90.7 | 77.6 | 84.6 | 52.4 | 42.8 | 33.0 | 48.4 |
| *QwQ-32B | 54.3 | 70.4 | 21.8 | 19.7 | 37.3 | 77.9 | 62.9 | 28.3 | 56.4 | 78.7 | 83.3 | 71.2 | 77.9 | 48.7 | 40.7 | 29.9 | 45.5 |
| Gemma3-27B | 40.7 | 76.0 | 10.7 | 5.6 | 30.8 | 61.1 | 24.7 | 6.6 | 30.8 | 75.0 | 83.0 | 67.5 | 75.2 | 28.7 | 21.9 | 15.0 | 26.1 |
| *DeepSeek R1 | 54.5 | 87.4 | 5.0 | 6.3 | 32.9 | 79.3 | 66.4 | 20.7 | 55.5 | 80.3 | 87.5 | 70.8 | 79.5 | 54.5 | 42.1 | 32.3 | 49.9 |
| Claude3.7 | 50.3 | 84.6 | 10.6 | 16.7 | 37.3 | 78.0 | 54.7 | 16.9 | 49.9 | 67.9 | 85.8 | 59.5 | 71.1 | 46.4 | 37.4 | 27.8 | 43.1 |
| *o4-mini | 57.7 | 86.6 | 8.1 | 17.1 | 37.3 | 86.2 | 70.9 | 19.5 | 58.9 | 78.1 | 86.6 | 66.8 | 77.2 | 57.0 | 57.4 | 46.5 | 57.4 |
| GPT-4o | 43.0 | 89.0 | 9.6 | 8.8 | 35.8 | 66.8 | 54.6 | 20.7 | 47.4 | 54.8 | 73.4 | 37.9 | 55.4 | 37.9 | 26.1 | 18.6 | 33.5 |
| GPT-4o-mini | 37.0 | 73.5 | 8.7 | 19.8 | 34.0 | 48.4 | 39.1 | 13.6 | 33.7 | 53.2 | 69.1 | 39.0 | 53.8 | 29.7 | 21.3 | 14.2 | 26.6 |

Table 1: Evaluation results of mainstream open-source and proprietary models on text-based tasks. Adopted metrics are usefulness (Use.), originality (Orig.), surprise (Surp.), and creativity (Crea.). Combined Creativity is the average creativity score of models on both convergent and divergent tasks, showing a model's comprehensive creativity. All scores are in $[0, 100]$. **Bold** numbers are models with the best corresponding performance, and the underlined ones are the corresponding second-best. (*) represents a reasoning model with thinking process.

| Model Name | Combined Creativity | Convergent Creativity | | | | Divergent Creativity | | | |
| --- | --- | --- | --- | --- | --- | --- | --- | --- | --- |
| | | Vision Question Answering | | | | Visual Story Telling | | | |
| | | Use. | Orig. | Surp. | Crea. | Use. | Orig. | Surp. | Crea. |
| Qwen2.5-VL-32B | 48.4 | 58.8 | 15.4 | 12.6 | 28.9 | 63.8 | 85.0 | 54.6 | 67.8 |
| Deepseek-VL2 | 46.3 | 66.3 | 16.6 | 18.8 | 33.9 | 53.4 | 79.4 | 43.0 | 58.6 |
| Qwen2.5-VL-72B | 42.0 | 54.4 | 14.1 | 10.2 | 26.2 | 48.6 | 79.2 | 45.4 | 57.8 |
| Intern3-VL-78B | 38.8 | 29.4 | 15.1 | 18.6 | 21.0 | 48.8 | 79.8 | 41.0 | 56.6 |
| Llama3.2-11B-V | 34.6 | 25.9 | 12.1 | 4.1 | 14.0 | 49.6 | 73.8 | 42.4 | 55.2 |
| Qwen2.5-VL-7B | 38.7 | 42.4 | 10.7 | 19.5 | 24.2 | 46.6 | 74.4 | 38.4 | 53.2 |
| Claude3.7 | 56.6 | 70.0 | 28.0 | 12.6 | 36.9 | 70.2 | 93.0 | 65.5 | 76.2 |
| o4-mini | 59.5 | 73.0 | 30.6 | 36.2 | 46.6 | 67.2 | 90.6 | 59.4 | 72.4 |
| Claude3.5 | 46.9 | 62.5 | 19.4 | 7.9 | 29.9 | 61.6 | 82.6 | 47.4 | 63.8 |
| GPT-4o-mini | 43.9 | 51.5 | 14.3 | 10.2 | 25.3 | 55.0 | 81.8 | 50.6 | 62.4 |

Table 2: Experimental results of VLMs on multimodal tasks. All scores are in $[0, 100]$. **Bold** and underlined fonts indicate the best and second-best performing models.

o4-mini models are evaluated in text-only and vision-language contexts (Menick et al., 2024; Hurst et al., 2024). Similarly, from Anthropic, we test Claude 3.5 Sonnet and Claude 3.7 Sonnet, both of which are also evaluated across both modalities to gauge their multimodal creative performance (Anthropic, 2024; 2025).

In the open-source domain, our selection is categorized by modality. For text-only tasks, we include Qwen2.5 (7B–72B; Instruct), Qwen3-8B (thinking and non-thinking), QwQ-32B (Team, 2024a; 2025), DeepSeek-R1 (Guo et al., 2025), and Gemma 3-27B (Team et al., 2025). Our open-source vision-language models (VLMs) consist of Qwen2.5-VL (7B, 32B, 72B; Instruct) (Team, 2024b), DeepSeek-VL2 (Lu et al., 2024a), InternVL3-78B (Zhu et al., 2025), and Llama-3.2-11B-Vision (Meta, 2024).

**Datasets.** Convergent tasks admit clear, verifiable answers or functional goals, whereas divergent tasks do not assume a single ground truth and emphasize novelty and appropriateness. For convergent creativity, we evaluate three kinds of tasks with ground truth: *Question Answering (QA)*, *Code Generation*, and *Visual Question Answering (VQA)* . For QA, we use **TriviaQA** (Joshi et al., 2017), which is a large-scale, evidence-grounded reading comprehension and open-domain QA from real queries. For Code Generation, we use **LiveCodeBench** (Jain et al., 2024) to provide program synthesis benchmarks with unit tests that enable objective and reproducible correctness checks. For VQA, we use **OK-VQA** (Marino et al., 2019), which pairs near-duplicate images to reduce language priors and foreground genuine visual understanding in QA. For divergent creativity, we evaluate on two different open-ended generation tasks, Story Telling and Idea Generation. For textual story telling, we utilize **WritingPrompts** (Hugging Face Datasets, 2024), which consists of prompt–story pairs for free-form narrative generation. For visual story telling, we adopt the stories introduced in

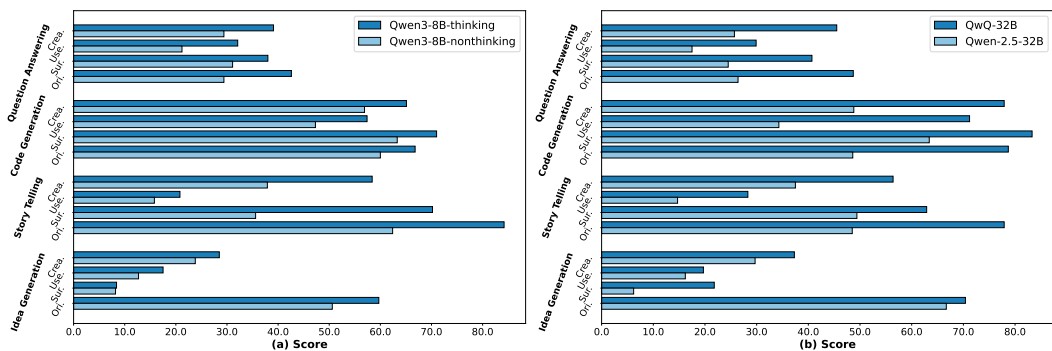

Figure 3: Reasoning vs. non-reasoning models of the same size. **Left (a):** Qwen3-8B in thinking mode versus Qwen3-8B in non-thinking mode. **Right (b):** Qwen2.5-32B vs QwQ32B.

**Creation-MMBench** (Fang et al., 2025) to assess image-grounded creative generation. We evaluate the idea generation capability on **MacGyver** (Tian et al., 2024), which presents constrained, real-world–inspired scenarios that require multi-step reasoning and inventive use of resources to achieve feasible solutions.

## 4.2 RESULTS AND DISCUSSIONS

### 4.2.1 MAIN RESULTS

**Which FM has the most creative mind?** For text-based tasks, we can see from Table 1 that *Qwen3-235B* and *o4-mini* are the most and the second most creative models. Qwen3-235B achieves the best performance on code generation and story telling, showing its expertise in long-tail creative content generation. While in QA and idea generation tasks, whose outputs are typically shorter, o4-mini performs better than other models. These results indicate that open-source LLMs can hold comparable and even better creativity than commercial LLMs. For vision-based tasks, our results summarized in Table 2 reveal that proprietary models currently possess the most creative capabilities. *o4-mini* achieves the highest combined creativity score (59.5), illustrating its remarkable talent for ingenuity within the structured constraints of the VQA task. In parallel, *Claude 3.7* demonstrates superior imaginative fluency, dominating the open-ended Visual Story Telling task. A clear creativity gap separates these leading proprietary models from the top open-source models. Among the open-source ones, *Qwen2.5-VL-32B* is the top performer due to its strong narrative generation, though it falls short in producing diverse solutions for constrained problems. The second-best open-source model, *Deepseek-VL2*, displays the opposite strength profile, excelling in convergent creativity while being less skilled in divergent story telling.

**Larger sizes do not always mean greater creativity.** Creativity does not increase monotonically with model size. In the U-O-S panel, larger models are not necessarily more creative. In the LLM-based evaluation, within the same model family, *Qwen2.5-32B* attains a combined creativity score of 35.4, slightly higher than the larger *Qwen2.5-72B* at 35.3; likewise, *Gemma3-27B* at 40.7 clearly exceeds the score of *Qwen2.5-72B*. In the VLM-based evaluation, this phenomenon is most clear within the open-source *Qwen2.5-VL* family, where the 32B variant (48.4) significantly outperforms its much larger 72B counterpart (42.0) in combined creativity. Similarly, the *Intern3-VL-78B* model is surpassed by the smaller 32B Qwen model. This suggests there may be an optimal balance in model size for fostering creativity, and simply increasing the parameter count is not a guaranteed path to a more creative machine mind.

**Reasoning yields larger marginal gains in creativity at scale.** Reasoning models often present impressive performances in creativity evaluation. For example,*Qwen3-8B-thinking*, *Qwen3-235B*, *QwQ-32B*, *DeepSeek R1* and *o4-mini* always outperform in creativity scores. Notably, scaling up the length of output tokens appears more important than scaling up the size of the model w.r.t. increasing FMs' creativity. As we can see in Figure 3, reasoning models (QwQ-32B,Qwen3-8B-thinking) significantly outperform non-reasoning models (Qwen32B,Qwen3-8B-nonthinking) under all creativity metrics, though they have the same model size.

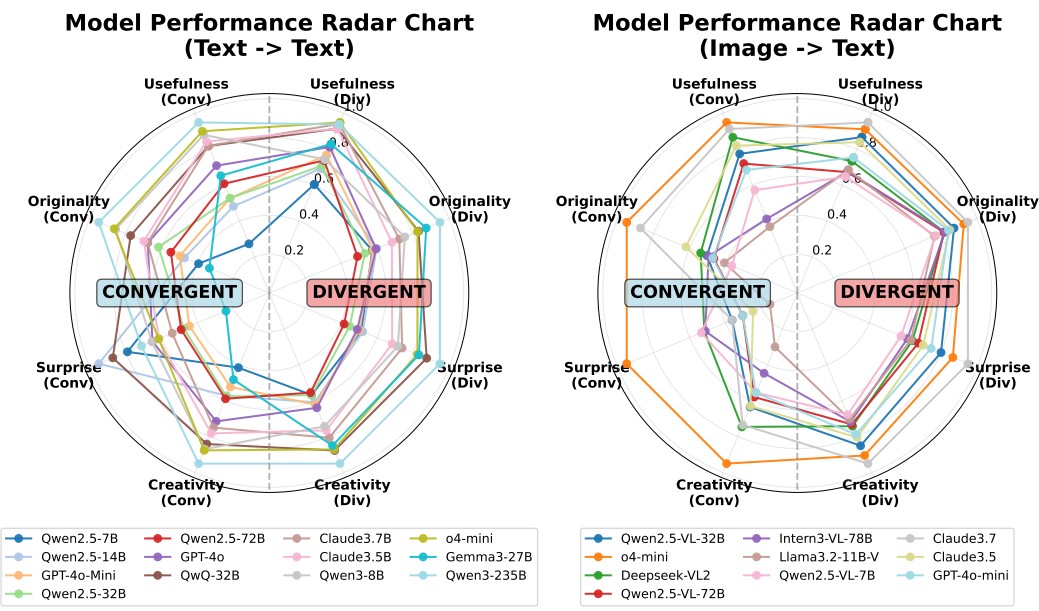

Figure 4: Radar chart illustrates the multi-dimensional components of creativity. The eight axes in each sub-figure represent four indicators (usefulness, originality, surprise, creativity) under both divergent (Div.) and convergent creativity (Conv.). All values here are normalized to (0,1).

### 4.2.2 CORRELATION BETWEEN DIVERGENT AND CONVERGENT CREATIVITY

To understand whether the capabilities that drive creative performance in structured problem-solving also translate to open-ended generation, we investigate the correlation between convergent and divergent creativity across different models and modalities and present two radar charts in Figure 4. For the text generation tasks (left radar chart), our analysis encompasses convergent tasks such as Question Answering (using TriviaQA) and Code Generation (using LiveCodeBench), as well as divergent tasks like Textual Story Telling (using WritingPrompts) and Idea Generation (using Mac-Gyver). For the visual language tasks (right radar chart), we analyze convergent Visual Question Answering (on OK-VQA) and divergent Visual Story Telling (on Creation-MMBench).

**Varied U-O-S Strengths Across Textual Tasks.** For the text to text generation tasks, the left radar chart of Figure 4 reveals that models often display varied strengths across the U-O-S dimensions when moving between convergent and divergent tasks. It illustrates that a model might exhibit high usefulness in structured, convergent tasks like code generation, but a greater capacity for originality in open-ended, divergent tasks such as story telling. This suggests that the mechanisms supporting creativity are not always uniformly leveraged across different task types, leading to distinct creative profiles and trade-offs in performance.

**Diverse Multimodal Creativity Profiles.** Similarly, the right radar chart of Figure 4, focusing on the image to text generation tasks, demonstrates a complex relationship between convergent and divergent creativity in multimodal contexts. Models show diverse performance distributions across the U-O-S components for tasks like visual question answering (convergent) and visual story telling (divergent). For instance, while *o4-mini* exhibits strong U-O-S performance in both regimes, *Claude3.7* demonstrates a particularly pronounced increase in U-O-S when generating visual stories, suggesting a stronger capacity for imaginative fluency in divergent creativity. This highlights that the balance between these creative attributes often varies depending on the specific task demands.

### 4.2.3 INFLUENCE OF CREATIVE INSTRUCTIONS

To understand how directive prompting can modulate a model's creative output across different task paradigms, we examine the influence of explicitly provided creative instructions on model perfor-

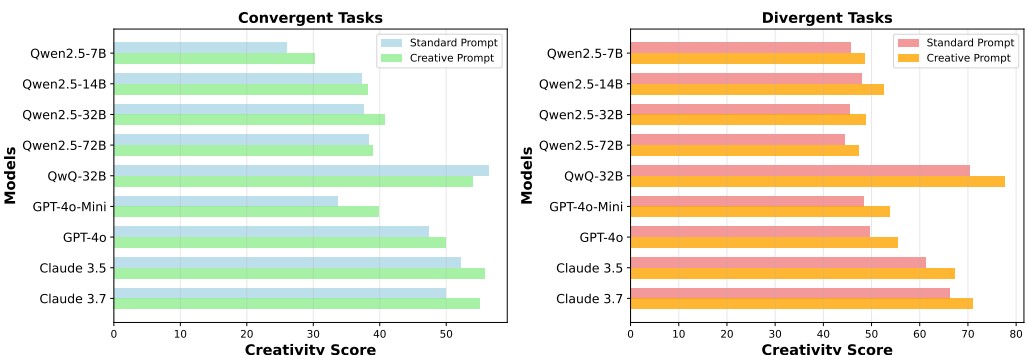

Figure 5: Ablation study on the effectiveness of creative instructions. We can see that adding creative instructions can generally boost both convergent and divergent creativity of FMs.

mance, comparing "Standard Prompts" with "Creative Prompts" which explicitly ask for creative content. For this analysis, we selected a diverse set of leading open-source and proprietary Foundation Models, encompassing various capacities and architectures to ensure broad representativeness, as detailed in Section 4.1. Specifically, for the convergent task, our analysis focuses on Code Generation using the LiveCodeBench dataset, while for evaluating divergent creativity, we utilize the WritingPrompts dataset for Textual Story Telling. The detailed instruction formats for these two prompts are provided in the Appendix B.

**Dual Impact of Creative Instructions in Convergent Tasks.** For convergent tasks, depicted in the left bar chart of Figure 5, models generally show an increase in their creativity scores when provided with the creative instruction, suggesting that explicit guidance towards creativity can help in generating more diverse or non-obvious correct solutions within constrained problem-solving contexts. However, a notable exception is that the reasoning model *QwQ-32B* exhibits a decreased creativity score with the creative instruction. This counter-intuitive result may indicate that for models highly optimized for logical reasoning and correctness in convergent tasks, an explicit creative instruction could inadvertently prompt a deviation from the most effective or accurate solution path. In tasks where *Usefulness* is synonymous with correctness, encouraging excessive novelty or surprise might lead to outputs that, while creative in intent, are less functionally optimal, thereby lowering the overall creativity score.

**Universal Benefit of Creative Instructions in Divergent Tasks.** Conversely, for divergent tasks, illustrated in the right bar chart of Figure 5, all models demonstrate an improvement in their creativity scores when given the creative instruction. This consistent positive effect highlights that creative instructions are highly aligned with the inherent open-ended nature of divergent tasks. Since these tasks prioritize exploration, perceived novelty, and unexpectedness in generated content, explicit prompts encouraging creativity allow models to fully leverage their generative capacities without the constraints of a single ground truth.

## 5 CONCLUSION

In this study, we introduce $C^2$-Eval, a comprehensive benchmark for systematically evaluating the creativity of Foundation Models. Our $C^2$-Eval establishes a convergent $\times$ divergent creativity ($C^2$) evaluation pipeline. Guided by creativity theories in social science, we adopt a unified U-O-S (usefulness, originality, surprise) triplet as the evaluation criterion, enabling comparable and diagnostic measurement of creativity across tasks and models. Through a systematic assessment of more than twenty open-source and proprietary FMs, we reveal several novel insights: Different from accuracy, creativity does not increase monotonically with model size; stronger reasoning ability and well-designed creative instructions reliably improve U-O-S; and convergent and divergent abilities are related, yet exhibit notable trade-offs, with creative instructions yielding particularly large gains in divergent settings. Overall, we hope $C^2$-Eval will encourage the community to examine the design of creativity benchmarks more critically and to develop more accurate and systematic evaluations.

ETHICS STATEMENT

This study evaluates foundation models solely along the dimension of creativity. All datasets used in $C^2$-Eval are publicly available and contain no personally identifiable or sensitive information. The benchmark measures generative outputs only, and should not be interpreted as evidence of underlying cognition. In addition, we employ LLMs as automatic judges for divergent tasks, and their assessments may inherit social or cultural biases from the underlying models. To mitigate risks, we (i) transparently report the limitations of the benchmark, and (ii) caution that $C^2$-Eval results should not be applied directly in high-stakes domains such as education, hiring, or healthcare without careful human oversight.

REPRODUCIBILITY STATEMENT

We place strong emphasis on reproducibility and provide full details of the benchmark design in the paper. All datasets used in $C^2$-Eval are public and properly cited. The evaluation pipeline—including automatic graders, rating prompts, and scoring rubrics—is described in detail in the methodology and appendix. We will release all code, evaluation scripts, and dataset splits under an open-source license, allowing researchers to fully reproduce our results. In addition, we will provide example scripts to facilitate running new models on $C^2$-Eval and to automatically generate the tables and figures reported in this paper, ensuring consistent results across different research teams.

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

APPENDIX

## A LLM USAGE STATEMENT

We utilized a Large Language Model (LLM) solely for the purpose of polishing the writing and improving the linguistic clarity of this manuscript. The LLM was used as a general-purpose assist tool for minor edits and stylistic refinements, and did not contribute to the ideation, research, methodology, or content generation of the paper. The authors take full responsibility for all content presented in this paper, and the use of the LLM does not imply authorship.

## B PROMPT TEMPLATES USED IN OUR TASKS

**Response Generations.** Below are the prompt templates in each step of $C^2$-Eval. We carefully designed each prompt to make sure that models can follow our instructs responsibly.

---

**Question Answering**

Answer the following question concisely. Output only the answer. {question}

---

**Visual Question Answering**

*<Image>*
*<Question>*

Answer the question about the image concisely. Provide only the main answer and your confidence score, without extra explanations.

Your output must strictly follow this format:
Answer: [Your Answer]

---

**Text Storytelling**

You are a writer. When given a prompt, write a short story only.

*<Story title>*

Rules:
- 500–900 words.
- Clear beginning–middle–end; coherent plot.
- Use concrete details; keep the prose readable.
- Output the story text only — no titles, notes, or meta-commentary.
- **Originality: avoid clichés and stock phrases; prefer fresh, specific imagery.**
- **Surprise: include one non-obvious turn that feels earned in hindsight.**
- **Voice: maintain a distinctive narrative voice throughout.**
- **Sensory depth: include at least three sensory details (sound, smell, texture, or precise objects).**

---

## Code Generation

You are an expert competitive programmer and algorithm specialist. Given a programming problem, provide a complete solution.

IMPORTANT REQUIREMENTS:

**- For LeetCode problems:**
Write complete code that reads input, parses it, calls the solution function, and prints the output.

**- For AtCoder/CodeForces problems:**
Write complete code with standard input/output handling using input() and print()
- Include ALL necessary imports
- Handle edge cases properly
- Make sure the code is directly executable
- DO NOT include any explanation or markdown formatting, just the code.

For AtCoder/CodeForces problems, follow this pattern:
------------------------------------------------------------------
``` python

# Read input
  n = int(input())
# ... read other inputs based on problem description

# Solution logic
# ... your solution

# Print output
print(result)
```

------------------------------------------------------------------
For LeetCode problems that require parsing:
------------------------------------------------------------------
``` python

# Parse input
import json
lines = []
while True:
    try:
        lines.append(input())
    except EOFError:
        break

# Parse the specific inputs
# parse based on problem requirements

# Define and call solution function
def solution(...):
    # your solution here
    return result

result = solution(...)
print(str(result).lower() \newline if isinstance(result, bool) else
    result)\newline
```

------------------------------------------------------------------
**Diversity Requirement:**
**Choose a distinct algorithmic idea or coding style from all previous attempts**
**(e.g., brute-force → hashing, recursion → iterative, DP → greedy).**
**Output ONLY executable Python code (no markdown, no commentary).**

---

### Visual Storytelling

You are a **talented** children's animation creator who have **a profound insight into the story plot and a unique understanding of the picture content, and have the ability to continue to narrate and complete a wonderful story** according to the content of the previous foreshadows.
*<Image List>*
**Background:** *<Instance-level Background>*
Please follow the requirements below to continue the story with a natural expansion of the plot or characters.
**Requirements:** 1. Ensure the story is clear, logically connected, and continues from existing content to form a complete, cohesive narrative; 2. Make the narration **vivid** and engaging through detailed descriptions and realistic dialogue; 3. Keep characters true to their original personalities while allowing for meaningful growth; 4. Introduce new, coherent challenges that naturally drive the story forward.

---

### LLM-as-a-Judge for Text-based Storytelling

Evaluate the SOLUTION to the PROBLEM under the following U-O-S rubric.
Scores are on a 0-5 scale and may be fractional (e.g., 3.5).
**- Usefulness**
Definition: The extent to which the story is logically coherent, well-structured, and effectively incorporates the elements of the writing prompt, providing meaningful or satisfying outcomes.
Scoring:
0-1: Disjointed, illogical, or barely related to the prompt.
1-2: Somewhat logical or connected but with major flaws.
2-3: Reasonably structured and adequately prompt-related.
3-4: Well-structured, strong logical flow, and integrates most prompt elements naturally.
4-5: Perfect logical flow, deeply connected to and creatively expanding upon the prompt.

**- Originality**
Definition: Degree to which the story presents novel, creative, or unexpected ideas that go beyond clichés. Scoring:
0-1: Story relies heavily on clichés or formulaic patterns.
1-2: Some novel elements present, but largely familiar.
2-3: Good originality with some fresh ideas.
3-4: Highly original elements noticeable.
4-5: Exceptionally original, offering new perspectives and highly imaginative developments.

**- Surprise**
Definition: The degree to which the story contains unexpected, non-obvious, or strikingly innovative elements that break conventional expectations and elicit an "aha" or emotional reaction. Scoring:
0-1: Entirely predictable, lacking any surprising or unexpected elements.
1-2: Mostly predictable with occasional minor surprising moments or details.
2-3: Noticeable surprising elements or twists at the detail level, adding some freshness without major disruption to the main storyline.
3-4: Several significant and creative surprises that meaningfully enhance the narrative's interest and originality.
4-5: Consistently delivers profound, unexpected developments that are highly surprising yet fitting, producing multiple `"aha"` moments throughout the story.
Return JSON with keys: `{"usefulness": number, "originality": number, "surprise": number, "overall": number, "notes": string}`.
The `"overall"` is a holistic judgment, NOT a simple average. Keep `"notes"` within 2 short sentences.

### LLM-as-a-Judge for Visual-based Storytelling

Please evaluate the creativity of the following story continuation, based on the provided sequence of images and accompanying writing instructions. Your task is to assess the continuation using the three creativity-related criteria defined below. You should integrate both the visual content from the images and the context provided by the prompt to inform your judgment. However, your scores should be grounded primarily in the criteria, with visual information serving as complementary context.

Evaluate the continuation along these three dimensions, giving each a score from 0 to 5:

**1. Usefulness**
Definition: How logically coherent, well-structured, and meaningfully aligned the continuation is with the visual and textual prompt.
Scoring:
- 0-1: Disconnected or incoherent.
- 1-2: Some relevance but contains logical flaws.
- 2-3: Reasonably coherent and contextually aligned.
- 3-4: Strong structure and natural integration of context.
- 4-5: Seamlessly extends the prompt and visuals.

**2. Originality**
Definition: Degree to which the continuation presents novel, creative, or unexpected ideas that go beyond clichés and predictable narrative paths.
Scoring:
- 0-1: Relies heavily on common tropes, lacks freshness.
- 1-2: Some creative elements, but mostly conventional.
- 2-3: Includes novel or imaginative concepts.
- 3-4: Clearly introduces distinctive, fresh ideas.
- 4-5: Exceptionally original and inventive.

**3. Surprise**
Definition: The degree to which the continuation includes unexpected, striking, or emotionally engaging developments.
Scoring:
- 0-1: Entirely predictable.
- 1-2: Mostly expected with minor surprises.
- 2-3: At least one interesting or innovative moment.
- 3-4: Several compelling or emotionally resonant twists.
- 4-5: Continuously surprising and deeply engaging.

**Output Format:** Please respond using the structure below:
Originality: $<Score>$
Usefulness: $<Score>$
Surprise: $<Score>$
Justification: $<$3-5 sentences explaining your evaluation, referring to relevant aspects of the story, prompt, and images. Avoid any commentary beyond the evaluation.$>$

Reply using ONLY this format, with no additional text.

# C CASE STUDY

## C.1 CREATIVE INSTRUCTIONS CAN HURT REASONING MODELS ON CONVERGENT CREATIVITY

---
**Problem Description**

Given an integer $k$, return the $k$-th character of an iteratively defined string: start with `"a"`; at each step append a "shifted" copy where each letter is advanced by 1 with wraparound ($'z' \rightarrow 'a'$). The length doubles each step, so efficient solutions must avoid explicit construction for large $k$.

---
**Response from QwQ-32B with and without the Creative Instruction**

**With Creative Instruction — Wrong:**

```python
import json
lines = []
while True:
    try:
        lines.append(input())
    except EOFError:
        break
k = int(lines[0])
def find_kth_char(k):
    word = 'a'
    while len(word) < k:
        next_str = []
        for c in word:
            if c == 'z':
                next_str.append('a')
            else:
                next_str.append(chr(ord(c) + 1))
        next_str = ''.join(next_str)
        word += next_str
    return word[k-1]
result = find_kth_char(k)
print(result)
```

---
**Without Creative Instruction — Correct:**

```python
k = int(input())
current_length = 1
while current_length < k:
    current_length *= 2
current_pos = k
shifts = 0
while current_length > 1:
    half = current_length // 2
    if current_pos > half:
        shifts += 1
        current_pos -= half
    current_length = half
result = chr(ord('a') + (shifts % 26))
print(f'"{result}"')
```

**Why the creative instruction hurts here.** The creative instruction biases QwQ-32B towards non-obvious constructions, which is creative in spirit, but algorithmically brittle: time and memory scale as $O(k)$. On large $k$, this deviates from the most reliable reasoning path, so **Usefulness** drops and the composite creativity score decreases, even if perceived surprise rises.

## C.2 RELIABILITY OF LLM-AS-A-JUDGE IN DIVERGENT TASKS

To make sure the evaluation results of LLM-as-a-Judge align with human preferences. We employ human-evaluations as the supplement to (V)LLM generated ratings. Here, we asked 10+ humans with solid educational backgrounds and brainstorm experiences to give their authentic assessments to each question-response pair with their justifications. Below are some examples from each subtask, and the comparison between the ratings given by (V)LLMs and human evaluations demonstrates the robustness of using advanced generative models as the evaluator.

| Question (Text Story Telling) | |
|---|---|
| You've finally managed to discover the secret to immortality . Suddenly , Death appears before you , hands you a business card , and says , When you realize living forever sucks , call this number , I 've got a job offer for you . | |
| **GPT-4o-mini** | **Qwen3-235B** |
| The day had dawned crisp and clear, the sun painting the sky a bright canvas of gold and azure. I stood atop a jagged cliff overlooking the restless sea, wind whipping my hair, feeling the raw pulse of existence thrumming beneath my skin. In my hand, I gripped a vial filled with a shimmering liquid — the culmination of years spent in forgotten libraries,. ………. | The vial didn't glow. It didn't hum with ancient power. It just sat there on the lab bench, a faint, iridescent sheen on its surface like oil on water, smelling sharply of ozone and burnt almonds—the byproduct of my twenty-year obsession colliding with a faulty capacitor. I'd done it. Telomeres repaired, cellular decay reversed, perfect metabolic stasis. ………. |
| **LLM Evaluation(gpt 5)** | |
| A **unclear, not readable** rendition of the classic "immortality is a curse" arc whose familiar …. **Originality: 2.8** **Usefulness: 3.6** **Surprise: 1.8** | **Vivid, tactile world-building** and the "recruitment, not release" reframe make the trope feel fresh…. **Originality: 3.9** **Usefulness: 4.7** **Surprise: 3.6** |
| **Human Evaluation** | |
| The story feels **formulaic** with a middling plot—very **ordinary**. It reads smoothly but **isn't engaging**; I **didn't want to read it**. **Originality: 2.0** **Usefulness: 3.2** **Surprise: 1.0** | There's one **nice place attracting me** and **the overall execution is solid**. I could finish it, **just not with much excitement. Originality: 3.0** **Usefulness: 4.0** **Surprise : 2.0** |

## Question (Idea Generation)

Your puppy chewed up a pillow, and now there are feathers everywhere. You don't have a vacuum cleaner available. Tools available include a yoga ball, a fan, a broom with a short, broken handle, a lint roller, and a roll of weak-adhesive packing tape. The tape can't pick up the feathers and the broom is difficult to use. How will you clean up the feathers using only these items?

| GPT-4o-mini | Qwen3-235B |
|---|---|
| Step 1: Use the yoga ball to gently roll over the feather-covered area, which will help to clump the feathers together. | Step1: Use the fan to blow feathers into a concentrated pile in a corner. |
| Step 2: Use the lint roller to pick up the clumped feathers. Roll it over the areas where the feathers have gathered. | Step2: Roll the yoga ball through the pile—the static charge from friction will attract and stick feathers to the ball. |
| Step 3: For any remaining feathers, use the weak-adhesive packing tape. Wrap a small piece around your hand, sticky side out, and pat it on the areas with leftover feathers. | Step3: Peel feathers off the ball by hand or use the lint roller for residual bits. |
| This method should effectively clean up the feathers without a vacuum. | *(Static electricity from the yoga ball effectively collects feathers; fan reduces scatter.)* |

## LLM Evaluation (gpt-5)

| | |
|---|---|
| Using the yoga ball to clump feathers is a fresh idea. But **relying on weak tape (stated ineffective)** and **not leveraging the fan undercuts feasibility and constraints.** | **Clever static use of the yoga ball plus fan-herding** fits the constraints. Minor practical risks (**insufficient static or over-scatter**) but likely workable. |
| **Usefulness: 2.0** 
 **Originality: 2.8** 
 **Surprise: 2.5** | **Usefulness: 3.6** 
 **Originality: 3.9** 
 **Surprise: 3.3** |

## Human Evaluation

| | |
|---|---|
| This solution is **practical** and **straightforward, using items in their obvious ways** (lint roller for picking up feathers, tape for small remnants, rolling the yoga ball to gather clumps). It works, but it mostly follows **conventional logic**. | This solution shows higher creativity because it introduces **an unconventional physical principle** (static electricity), it **leverages multiple items in novel ways** (fan to corral feathers, yoga ball to generate static for collection). It **goes beyond standard** "use item for its intended purpose" thinking. |
| **Usefulness: 3.5** 
 **Originality: 2.2** 
 **Surprise: 2.0** | **Usefulness: 4.2** 
 **Originality: 3.8** 
 **Surprise: 4.0** |

| **Question (Visual Story Telling)** | |
|---|---|
| 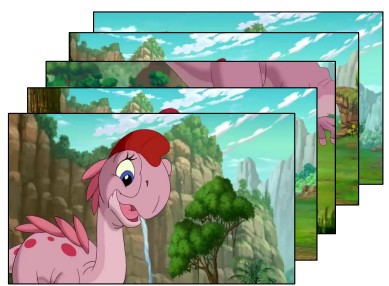 | *You are a talented children's animation creator who have a profound insight into the story plot and a unique understanding of the picture content, and have the ability to continue to narrate and complete a wonderful story according to the content of the previous foreshadows. Please follow the requirements below to continue the story with ......* |
| **Qwen2.5-VL-7B** | **OpenAI-o4-mini** |
| Breezy breeze, the little pink dragon with the sparkling blue eyes, watched as her tiny friend, the yellow dino, made his first attempt at standing up on his hind legs. His curiosity was clear, and his eyes sparkled with excitement. "Wow, Breezy, look at me! I did it!" the yellow dino exclaimed with a triumphant smile, a little dirt and mud on his tummy. He stood tall on the stone, feeling the earth beneath his paws. "Breezy, why are you standing back?"...... | That afternoon, as the sun climbed higher over the fern-blanketed valley, Pink and Yellow were practicing Yellow's new balance routine on the smooth, mossy stone. Pink danced lightly around Yellow, encouraging each tentative step. "Keep your head up, Yellow," Pink chirped, tapping her toe in time with the distant waterfall's roar. "Imagine you're walking along a leafy branch—steady and calm." Yellow swallowed hard, blinked, then planted one foot...then the other...... |

**LLM Evaluation (GPT-4o)**

| | |
|---|---|
| **Usefulness:** 2.9 
 **Originality:** 2.2 
 **Surprise:** 1.3 
 **Justification:** The story continuation is moderately creative. However, the concept of characters gaining confidence and exploring the world is fairly conventional and lacks significant novelty ...... | **Usefulness:** 4.8 
 **Originality:** 4.3 
 **Surprise:** 3.6 
 **Justification:** The continuation demonstrates originality by incorporating a rescue mission involving a baby Pteranodon, which adds depth to the characters' relationship and highlights their growth ...... |

**Human Evaluation**

| | |
|---|---|
| **Usefulness:** 2.3 
 **Originality:** 2,8 
 **Surprise:** 0.7 
 **Justification:** The story is logically coherent and well-structured. However, it relies heavily on predictable tropes, without introducing any novel concepts or unexpected plot developments. ...... | **Usefulness:** 4.9 
 **Originality:** 3.8 
 **Surprise:** 3.4 
 **Justification:** This story is exceptionally useful. It introduces a compelling external conflict that logically tests the characters' bravery ...... |

## D    LIMITATIONS AND FUTURE WORK

Our study opens up several avenues for future research. First, while $C^2$-Eval currently focuses on text and vision–language tasks, it does not yet extend to domains such as mathematics, nor to modalities like speech, video, or embodied interaction. Expanding the benchmark to these areas represents a natural and promising next step. Second, for divergent tasks we rely on large language models as automatic judges. Although explicit rubrics and consistency checks help guide the evaluation, the outcomes may still reflect biases inherent to these models. Future work should explore richer human annotations, cross-cultural perspectives, and adversarially constructed test sets to enhance robustness. Finally, our present analysis emphasizes system-level performance, leaving open questions about the mechanisms through which creativity emerges, whether from decoding strategies, dataset composition, or fine-tuning objectives. Investigating these finer-grained drivers of creativity will be an important direction for deepening our understanding of machine creativity.

