# OpenReview forum: "What Shapes a Creative Machine Mind? Comprehensively Benchmarking Creativity in Foundation Models"
_ICLR.cc/2026/Conference — Submitted to ICLR 2026_

### Official Review · Reviewer_H7Fy · 2025-10-27

**Soundness:** 2
**Presentation:** 3
**Contribution:** 2
**Rating:** 2
**Confidence:** 4

**Summary:**

The submission introduces the C2-Eval framework to assess the creative potential of various LLMs. The C2 comes from evaluating models on both "convergent" and "divergent" tasks. Convergent tasks are ones that can be evaluated for accuracy, whereas divergent are more open-ended (e.g., creative writing).
For either convergent and divergent tasks, the work proposes three metrics of Usefulness, Originality and Surprise. For convergent tasks, they're based on similarity between a set of responses, and/or model self-reported confidence. For divergent tasks, the work relies on LLM as a judge to evaluate U, O and S.

The findings indicate that different models shine at convergent or divergent creativity, with thinking models outperforming non-thinking models. Prompt engineering to encourage creativity helps almost all models on the benchmark.

**Strengths:**

- The paper is very well written and the motivation of wanting to study creativity potential in LLMs is strong.
- The experiments are thorough in that they involve a lot of different LLMs, both open- and closed-source, and involve varied tasks.

**Weaknesses:**

- The way the metrics are defined feels a bit ad-hoc, and there's no validation with expert annotation. For example focusing on the convergent tasks: usefulness is not really relating to creativity (just accuracy), originality is based on essentially set-wise diversity (which is somewhat unrelated) and surprise is essentially self-assessed by the LLM itself (though prior work has shown that asking an LLM for confidence often does not work). On the other hand for the divergent tasks, it is simply LLM-as-a-judge. But we would need some validation to trust that this protocol works. Given two samples that are judged to be more/less creative by this protocol, how likely is a human expert to agree?
- It feels to me that there are simple ways to modify the generation procedure that would lead in terms of these metrics to increased numbers. For instance increasing temperature, or over-generating and filtering similar responses. It is unclear however that this would in fact represent more creativity on behalf of the model, which shows the potential limitations of the metric.
- Regarding the use of LLM-as-a-judge: since we are evaluating whether LLMs can be creative, it feels like asking an LLM to do the judgement of whether something is creative assumes that the LLM understands creativity in the first place, which feels circular.
- Originality typically relates to something that has not been done before (by anyone else), yet here it seems to be defined more in relation to what an LLM has not done or generated before. Isn't that intrinsically limited?
- In general, can we work on creativity and creativity evaluation without involving creative professionals or participants that can judge the creativity of an artifact?

**Questions:**

Please see the listed questions in the Weaknesses section.

---

### Official Review · Reviewer_8wrJ · 2025-10-28

**Soundness:** 3
**Presentation:** 3
**Contribution:** 3
**Rating:** 4
**Confidence:** 4

**Summary:**

Attempts to operationalize a framework from the social sciences (e.g., Simonton 2012) to measure the creativity of foundation models undertaking various tasks. The authors split these tasks into those that require "convergent creativity" (e.g., question answering and code generation) and those requiring "divergent creativity" (e.g, storytelling) with entirely different metrics for each type. The framework measures creativity in terms of "usefulness", "originality", and "surprise".

**Strengths:**

The paper is enjoyable and fun to read. When evaluating the creativity of foundational models, it's important to look at it from a traditional social science perspective. The experiments are comprehensive and clearly reported.

**Weaknesses:**

I am not 100% happy about how the standard components of computational creativity are mapped into the benchmark. "Usefulness" is sometimes called "familiarity" or "appropriateness". Random nonsense is not creative, even if it is novel and surprising, because it seems to have no value and does not connect with the observer. I guess I can accept that we can reasonably measure usefulness in terms of correctness, coherence, etc.

My concern comes with the measures of originality and surprise. In the creative sense, they are related measures, differing by context. Originality is set in the broader social context. Is an artefact novel in the global sense that nobody or nothing has created something like this before? Surprise requires an observer. Is the artefact novel in the mind of the viewer? Or at least this is how I understand the distinction, having read the relevant literature, but long ago. To make a distinction, wouldn't an LLM-judge need to be grounded in the perspective of a particular observer (or class of observer) in order to make a distinction.

The distinction between originality and surprise is critical, and this is not really called out in the paper, which is very vague about the distinction. Moreover, there are just some vague sentences about how the distinction is operationalized.

It's also unclear which model is used as a judge. I see only that "we employ an automatic rater, generally the most advanced judge model". I've read the paper several times, but I don't see anything beyond that statement. I see lots of models being evaluated. Wouldn't it make sense to have each model in turn used as a judge.

**Questions:**

Convince me that your operationalizations of originality and surprise really align with standard models of creativity.

Why not have the models judge each other, rather than picking a single model? It's also unclear which model you are using for this purpose.

Is this UOS model really Simonton? I suppose I should go back and look, but that doesn't align with my memory.

---

### Official Review · Reviewer_YEZ2 · 2025-10-31

**Soundness:** 2
**Presentation:** 3
**Contribution:** 3
**Rating:** 4
**Confidence:** 4

**Summary:**

This paper develops C2-Eval, a benchmark to quantify creativity in foundation models (FMs) across both ‘convergent’ tasks (with objective correctness, e.g., QA, code) and ‘divergent’ tasks (open-ended generation, e.g., storytelling). The framework operationalizes creativity using a Usefulness–Originality–Surprise (U-O-S) rubric originated in social science and applies a scoring pipeline across task types, using LLM judges. The authors benchmark FMs and show that, interestingly, creativity does not scale monotonically with model size. Instead, increasing size of output tokens, using reasoning-based model and creative prompting yield disproportionately higher creative gains. Results reveal nuanced trade-offs between convergent and divergent creativity, indicating these capabilities are correlated but not interchangeable.

**Strengths:**

Timely contribution to the recent area of LLM creativity, with a new framework to measure both convergent and divergent creativity.

The proposed evaluation framework could be a good resource for the community to improve LLMs on creativity.

Experiments are rather extensive, evaluating several current LLMs on the proposed creativity measurement.

**Weaknesses:**

Framing tasks as mutually exclusive divergent and convergent might not be appropriate, for example, code generation is considered a convergent task here but there may be multiple possible code solutions for a given coding task.

Formulation of creativity measurement is questionable. For example, the final composite creativity score is an average across the usefulness, originality and surprise score values but not evidence to support this design choice. For example, a solution that is not useful may still score high on originality and surprise but will not have any practical value.

**Questions:**

In Figure 1, isn’t the autograder an LLM judge? If so, why is it depict to be different entities?

It would be interesting to see exactly why approaches such using reasoning-based model and creative prompting increases creativity, i.e. which of U-O-S is it improving.


Other related work on measuring divergent and convergent creativity that would be helpful to also discuss:
https://arxiv.org/abs/2407.09007
https://arxiv.org/pdf/2410.04197
https://openreview.net/forum?id=hocpzqMqB5
https://arxiv.org/abs/2507.18368

---

### Official Review · Reviewer_cK1g · 2025-11-01

**Soundness:** 1
**Presentation:** 2
**Contribution:** 1
**Rating:** 2
**Confidence:** 5

**Summary:**

This paper introduces C²-Eval, a benchmark for evaluating creativity in foundation models. The framework separates "convergent creativity" (structured tasks like code generation and QA) and "divergent creativity" (open-ended tasks like story generation), evaluating both through the lens of Usefulness, Originality, and Surprise (U-O-S).

**Strengths:**

No Strengths IMHO

**Weaknesses:**

The abstract threw me off. However, existing evaluation frameworks for creativity remain fragmented, relying on ad hoc metrics that are not firmly grounded in established theories of creativity. Just because authors knowledge of Creativity Research is limited doesnt mean work doesnt exist

- Art or Artifice? Large Language Models and the False Promise of Creativity (https://dl.acm.org/doi/10.1145/3613904.3642731)
- Human Creativity in the Age of LLMs: Randomized Experiments on Divergent and Convergent Thinking (https://dl.acm.org/doi/abs/10.1145/3706598.3714198)
- LLM Discussion: Enhancing the Creativity of Large Language Models via Discussion Framework and Role-Play (https://arxiv.org/abs/2405.06373)

This is a below average paper. The core components of this work are largely derivative.

- Standard benchmarks such as (TriviaQA, LiveCodeBench.) relabeled as "creativity" tasks
- There are 50 papers in HCI literature on Convergent and Divergent Thinking. Authors are reinventing the wheel
- The categorization of Question Answering as a creativity task is fundamentally problematic.  Factual QA requires correctness, not creativity
- Why U-O-S framework ? Why not other Creativity frameworks like Torrance Test or Consensual Assessment Technique ?
- For divergent tasks, using GPT-based judges is meaningless. GPT-4 judging GPT-4o's creativity is deeply flawed given its been shown LLMs favor their own generation.
- Inter-annotator agreement statistics are absent
- Lower confidence doesn't imply creative surprise. Its quite a leap of faith
- Temperature and sampling parameters unreported that is critical for generation diversity

**Questions:**

NA

---

### Meta-Review · Area_Chair_GCmC · 2026-01-24

**Summary:**

Reviewers agree the paper is clearly written and runs extensive experiments, but they remain unconvinced that C2-Eval actually measures “creativity” rather than proxies (accuracy, sampling diversity, and judge bias). A major theme is weak construct validity—especially for originality/surprise—and overstated novelty/positioning relative to existing creativity and HCI literature. Overall sentiment trends below the acceptance bar.

**Reviewer Concerns:**

Partially addressed:

* Missing implementation details (judge identity, judge prompt, generation settings like temperature/top-p, number of samples).

* Confusion about “autograder vs LLM judge” depiction.

* Rationale for aggregating U/O/S into a single composite score.

Still outstanding / central concerns:

* Metric validity: “Usefulness=correctness” is capability, not creativity; originality as embedding-cluster count is sampling- and representation-sensitive; “surprise = 1 − confidence” is a weak proxy (uncertainty ≠ creative surprise).

* LLM-as-judge limitations: circularity, self-preference/style bias, unclear mitigation; examples are not a substitute for systematic human/expert validation (and inter-annotator agreement).

* Framing/novelty: QA as a creativity task is disputed; convergent/divergent split may be too rigid; related work coverage and claims of “gap” are viewed as overstated.

**Reviewer Scores:**

* cK1g (2): likely stays 2.

* YEZ2 (4): likely stays 4 (could rise only with strong clarifications/justification).

* 8wrJ (4): likely stays 4.

* H7Fy (2): could move to 3 if robustness/validation is substantially strengthened; otherwise 2.

---

### Decision · Program_Chairs · 2026-01-26

Reject